# Effect of a Boric Acid Corrosive Environment on the Microstructure and Properties of Concrete

**DOI:** 10.3390/ma13215036

**Published:** 2020-11-08

**Authors:** Yu Wang, Bei Huang, Zhongyang Mao, Min Deng, Huan Cao

**Affiliations:** 1College of Materials Science and Engineering, Nanjing Tech University, Nanjing 211800, China; 201861203018@njtech.edu.cn (Y.W.); mzy@njtech.edu.cn (Z.M.); dengmin@njtech.edu.cn (M.D.); 201961203093@njtech.edu.cn (H.C.); 2State Key Laboratory of Materials-Oriented Chemical Engineering, Nanjing Tech University, Nanjing 211800, China

**Keywords:** nuclear safety, boric acid, concrete durability, properties, microstructure

## Abstract

Boric acid, a weak acid, is often used to shield neutrons in water cooling systems in nuclear power stations. The leakage of boric acid in water cooling systems damages the concrete structure and affects the safety of nuclear power engineering. In this experiment, concrete specimens were cured with boric acid at 20, 40, and 70 °C to study the effect of boric acid on the microstructure and properties of concrete. X-ray diffraction (XRD) and thermogravimetry and differential scanning calorimetry (TG-DSC) were used to analyze the change in mineral composition. The microstructure was examined by scanning electron microscope (SEM). The porosity of the concrete was examined by mercury intrusion porosimetry (MIP). The results show that the performance of specimens was stable under the curing conditions of 20 and 40 °C. Under the curing environment of 70 °C, the performance of concrete cured with 0, 2000, and 7000 ppm concentrations was stable, but the compressive strength of the 180,000 ppm specimen was reduced by 27.8% and suffered the most serious loss of mass and surface corrosion, with the most harmful pores. The high concentration of boric acid seriously damaged the surface structure of concrete, which is the main reason for its loss of properties. This situation is extremely dangerous in nuclear power engineering, so the effect of boric acid leakage cannot be ignored.

## 1. Introduction

In containment spray systems at nuclear power plants, boric acid is used to cool the containment, reduce pressure, and remove fission products. However, when abnormal leakage of boric acid in the containment occurs, the concentration of boric acid rises and approaches saturation as a result of water evaporation. The pH of boric acid solution at 95 °C is less than 3 [1], and it is relatively corrosive at a high temperature [2], with the ability to corrode stainless steel and concrete. In 2002, boric acid leakage occurred on the top of the reactor cooling tower of the DAVIS-Besse nuclear power plant in the United States, leaving less than half an inch of the original 6-inch-thick stainless steel alloy cap, which was subjected to more than 2200 pounds of pressure per square inch of area [3,4]. A boric acid water tank in a power station in China has been leaking since 2009 as a result of defects in the weld line of the inner steel lining. By 2017, the leaking liquid filled the interlayer between the water tank and the concrete wall [5]. There are many reports of similar incidents [6,7,8,9]; therefore, the leakage of boric acid in nuclear power plants and its significant impact cannot be ignored.

The corrosion by boric acid is one of the factors that leads to the lack of durability of concrete. Boric acid also has a neutral reaction with concrete. At room temperature, boric acid has no obvious effect on the compressive strength, flexural strength, wrapping strength, and corrosion rate of reinforcement of concrete specimens [10,11]. However, when nuclear waste containing boric acid is added to cement as an admixture, the compressive strength of cement decreases with the increase of nuclear waste content over a period of 90 days [12]. After sampling a nuclear power plant with boric acid leakage, X-ray diffraction (XRD) phase analysis showed that boric acid reacts with Ca(OH)_2_ in concrete and CaO in calcium silicate to generate various borate products and change various mineral components in cement [5]. Petrological inspection of concrete cylinder specimens eroded by boric acid showed that the cement paste under boric acid leaching was very soft and porous, with low inherent strength, and the cement mineral had an acid neutralization ability, which reduced the leaching rate of boric acid solution diffusion in concrete [13,14].

In the actual operation of the containment cooling tank, the normal temperature is 70 °C, and the concentration of boric acid is 7000–7700 ppm [15]. Generally, in the corrosion reaction of concrete, temperature tends to exacerbate the corrosion reaction, thus reducing the durability of concrete material [16]. In addition to temperature, acid corrosion of concrete is often related to pH and its permeability [17,18].

The deterioration of concrete is mainly due to the reduction of macroscopic mechanical properties, and the reason for the deterioration is the change of concrete structure. At present, the mechanism of deterioration of concrete is mainly analyzed by means of SEM and XRD [19,20]. Because the pore structure is one of the main factors affecting the strength of concrete, it is very important to study the change of pores in concrete during the corrosion process [21].

In the existing research, only the effect of boric acid on cement concrete at 20 °C has been studied, without taking into account the effect of temperature. Therefore, this study used the actual situation of a nuclear power plant as a simulation and the concentration of boric acid was set considering the polycondensation of boric acid in the limit case (70 °C). Thus, a batch of concrete specimens was formed, and an accelerated curing box was used to simulate the cooling water tank with curing temperatures of 40 and 70 °C. Concrete specimens cured with saturated acid concentrations were used to study the effect of boric acid on the microstructure and properties of cement concrete materials.

## 2. Materials

### 2.1. Raw Materials and Preparation of Cement

P.II 52.5 Portland cement (PC) was supplied by Onada Cement Corp in Nanjing, China. The fly ash (FA) used in this study was a byproduct produced by Huaneng Power Plant in Nanjing, China. This FA had a low CaO content of 3.40%, as shown in Table 1. The boric acid used in this study was nuclear industry standard from Guangdong Fine Chemical Engineering Technology (Guangdong, China), and its purity was above 99.99%. The fine aggregate was local river sand, the fineness modulus was 2.7, and grading was good for the medium sand. The coarse aggregate was made of two kinds of gravel with different particle sizes of 5–16 and 16–31.5 mm, and the two sizes accounted for 50% of the gravel. The water reducer adopted was PCA^R^-SCC high-performance concrete water reducer from Sobute New Materials Company, Nanjing, China.

### 2.2. Concrete Size and Mix Proportion

The concrete specimens consisted of Portland cement, fly ash, coarse aggregate, fine aggregate, and water reducer. The size of the specimens was 100 mm × 100 mm × 100 mm, and the microstructure and properties of cement concrete were measured. After 24 h of molding, all specimens were cured at 20 °C for 28 days. The specimens were placed in a curing box with a humidity of 95% and a temperature of 25 °C. The concrete used in a nuclear power plant has excellent performance, so we adopted the ratio of R55. The composition of the concrete specimens is shown in Table 2.

### 2.3. Boric Acid Curing Stage

A concrete accelerated curing box was used to simulate the actual temperature. In this experiment, we used 20, 40, and 70 °C as a comparative test. The concentrations of boric acid in the curing solution were 0 and 50,000 ppm at 20 °C; 0 and 90,000 ppm at 40 °C; and 0, 2000, 7000, and 180,000 ppm at 70 °C. These conditions are denoted by 20–0, 20–50,000, 40–0, 40–90,000, 70–0, 70–20,000, 70–7000, and 70–180,000 ppm for short. The percentage of boric acid and the concentration of solution were chosen according to the national standard. Specimens were acid-cured in the aforementioned environments, The curing of the specimen is shown in Figure 1.

The volume of boric acid was five times the volume of the specimen. The pH of the solution of boric acid at 70 °C (180,000 ppm) was about 4.2, and the pH of the solution did not change significantly within a week. Mannitol was used to measure the concentration. We changed the curing solution every other week to ensure the concentration of boric acid.

## 3. Methods

### 3.1. Mass Change and Loss Rate of Compressive Strength

The concrete samples were removed at the ages of 0, 28, 90, and 180 days, and the mass change of the specimens was measured using an electronic balance (JSC-1000C, accurate to within 0.1 g, Aipu, Changzou, Chian). The specimen was taken out of the curing box and dried for the same drying time for each age.

The compressive strength of concrete was calculated in strict accordance with the standard test method of mechanical properties of ordinary concrete using a compressive-testing machine with a loading rate of 0.5–0.8 MPa/s, and the calculated result was accurate to within 0.1 MPa. The compressive strength was measured according to standard [22]. The machine for testing mass and compressive strength is shown in Figure 2.

### 3.2. Depth of Neutralization

After reaching the required age, the concrete specimen was sliced about 1 cm from the surface, and then phenolphthalide alcohol solution with a mass fraction of 1% was used to test the depth of neutralization. Ten points were taken for each edge, and the average depth of neutralization was calculated.

### 3.3. XRD and TG-DSC Analysis of Mortar

Mortar was subjected to X-ray diffraction (XRD) and thermogravimetry and differential scanning calorimetry (TG-DSC) analyses at different ages. The mortar cut from the concrete samples was immersed in absolute ethyl alcohol for 12 h to terminate hydration, before being vacuum dried at 60 °C for 24 h and then ground into powder. The test powders were passed through an 0.08 mm sieve. The XRD data were collected in the range of 5° to 80°, 2θ at a counting time of 15 s/step, and a divergence slit of 1°. TG-DSC data were obtained from 50 to 1000 °C at a rate of temperature increase of 10 °C/min in an N_2_ atmosphere.

### 3.4. Pore Structure of Concrete

Mercury intrusion porosimetry (MIP) is a method that can accurately measure the porosity and pore size distribution of the cement material. In this study, we used MIP (AutoPore W9500, Micromeritics, Norcross, GE, USA) to characterize the porosity and pore distribution of the sample, and analyzed the corrosion process of the material from the pore structure.

### 3.5. Microstructure of Concrete

In order to investigate the morphology of concrete under boric acid attack, a characteristic fresh piece of concrete was sampled, hydration terminated, and vacuum dried. Then, the dried sample was embedded in a low-modulus epoxy, coated with a gold–palladium coating. A scanning electron microscope (JSM-6510, JEOL, Tokyo, Japan) was used to observe the surface morphology of the concrete.

## 4. Results and Discussion

### 4.1. Mass and Compressive Strength

The surfaces of the concrete samples at 90 days are shown above in Figure 3. The surface of specimen 70–0 ppm at 90 days was ash black, similar to that at 0 days. When the concentration of boric acid increased, the black color of its surface turned lighter and lighter. Surface desertification can be clearly observed for the 70–180,000 ppm sample.

The mass loss of the concrete samples is shown in Figure 4a. Under the curing condition of boric acid saturation concentration, the mass loss of specimens at 20 and 40 °C between 0 and 90 days was within the normal range. However, at 180 days, the mass loss of 40–90,000 ppm was 0.32%, more than double the mass loss rate of 20–50,000, which was 0.14%. The mass loss rates of 70–180,000 ppm at 28, 90, and 180 days were 0.06%, 0.57%, and 0.68%, and the mass loss rates of 70–180,000 ppm in each phase were far higher than those of other test pieces. Most importantly, the weight loss of a specimen reached 19.6 g at 180 days (the initial weight was 2470.8 g).

We studied the effect of the concentration of boric acid, as shown in Figure 4b. Under the conditions of 70 °C and 0, 2000, and 7000 ppm, the mass loss of specimens over time showed a steady trend. The mass loss rate of the 180 days and 70–70,000 ppm sample was only 0.15%, much lower than that of 70–180,000 ppm at 90 days. The comparison shows that the 70–18,000 ppm sample lost a considerable amount of weight in each phase, which was perfectly demonstrated by the desertification on the surface of the concrete sample. The set cement in the concrete was dissolved in boric acid, resulting in the serious weight loss of material.

Figure 5 shows the compressive strength development for concrete specimens under different acid conditions. The compressive strength of concrete was 64.4 MPa at 0 days, so this was considered as the initial strength of all specimens. At a temperature of 20 °C for conservation, the compressive strength of 20–0 and 20–50,000 ppm samples was 68.3 and 67.9 MPa at 28 days, respectively, indicating that the initial compressive strength of concrete consistently changed at room temperature, but the changing rules of compressive strength were different between 28 and 90 days. The compressive strength of 20–0 ppm was 78.8 MPa at 90 days, while that of 20–50,000 ppm was only 68.2 MPa, which was basically the same as the compressive strength of 20–50,000 ppm at 28 days. The growth of the compressive strength of concrete was hindered by boric acid between 28 and 90 days. At 180 days, the compressive strength of 20–0 ppm was 81.3 MPa, while that of 20–50,000 ppm was 69.7 MPa.

The change of compressive strength of 40–0 was basically the same as 20–0, but the difference between 40–90,000 and 20–50,000 ppm was most significant at 90–180 days; the compressive strength began to decline, and at 180 days, the compressive strength of 63.3 MPa was lower than the initial strength, and the loss rate of compressive strength reached 1.70%.

At a temperature of 70 °C for conservation, the compressive strength of 70–0, 70–2000, and 70–7000 ppm was 76.7, 78.2, and 75.6 MPa at 28 days, respectively. The compressive strength of material at 70 °C was greater than that at 20 °C, but the compressive strength of 70–180,000 ppm at 28 days was only 68.1 MPa, and the compressive strength of three other groups started to decrease at 28 days, which indicated that the growth of the compressive strength of concrete was hindered by highly concentrated boric acid in the early stage of conservation at a temperature of 70 °C. The compressive strength of 70–0, 70–2000, and 70–7000 ppm slightly decreased as concentration increased at 90 days, but that of 70–180,000 ppm was only 48.1 MPa at 90 days. The loss rate of compressive strength reached 25.3%, and the loss rate of compressive strength also reached 27.8% at 180 days.

In the process of measuring the compressive strength of this group of specimens, we also found that the failure modes of the specimens in this group were different from those in the other seven groups; the failure mode of 70–180,000 ppm was found as shown in Figure 6a. The failure mode is mainly surface peeling and other groups as shown in Figure 6b, which shows that boric acid corroded concrete from the side to the surface and then to the inside.

### 4.2. Depth of Neutralization

In Figure 7, among the six groups of specimens, the depth of neutralization of 20–0, 20–50,000, 70–0, and 70–2000 ppm at 180 days was 0 mm. For 70–7000 ppm, the three sides (top, bottom, and left) of the specimen also had a depth of neutralization of 0 mm; the depth of neutralization on the right was 0.4 mm, so the depth of neutralization of the 70–7000 group was 0.1 mm. The depth of neutralization of 40–90,000 ppm was 0.2 mm. For 70–180,000 ppm, each edge had an obvious neutralization trace, and after measurement, the neutralization depth of 70–180,000 ppm was 1.5 mm.

The neutralization depth of 70–180,000 ppm was 1.5 mm; this proved that boric acid corroded concrete from the outside to the inside under a high temperature and high curing concentration environment. The basic cement was gradually dissolved out of the concrete by boric acid, and boric acid destroyed the surface structure of the concrete, making it brittle and reducing its compressive strength. This corresponds to the previous mass loss rate of concrete; in other words, except for the concrete itself, the weight loss was almost due to the boric acid corrosion of the surface layer of cement.

### 4.3. Mineral Composition Examination

In Figure 8, Ca(BO_2_)_2_ was not demonstrated in the XRD atlas; therefore, it was believed that concrete was generally corroded by boric acid in an alkaline environment. Under this condition, the major borate produced after concrete was corroded by boric acid was calcium metaborate. The concrete suffered from significant weight loss. Hence, during neutralization, boric acid firstly reacted with alkaline composition in paste to produce Ca(BO_2_)_2_, but as boric acid continued to corrode, the pH value continuously decreased, and the produced borate was slowly dissolved in the conservation solution of boric acid. The contents of Ca(BO_2_)_2_ and other borates in paste were extremely low, which was why they could not be detected during XRD.

The surface of the concrete showed an obvious sand formation at 90 days in Figure 9, so the surface of concrete was sampled, and the visible aggregate was removed as much as possible.

As age increased, the sand on the concrete surface is exposed; the peak of sand becomes much stronger than the rest, making it inconvenient to detect the rest. In order to observe the reaction products, we removed the peaks of cement and sand in the XRD; a group of peaks of Ca(BO_2_)_2_ appeared on the surface of the concrete at 180 days. Undeniably, the presence of this group of peaks confirms that boric acid generates Ca(BO_2_)_2_ in the alkaline environment of concrete. Because the concrete structure is not dense with cement pastes, the corrosion degree of concrete was more intense than others, so the XRD pattern is different from that of the other experimental group.

In order to study the change of content of calcium hydroxide (CH%) on the surface of the material, samples on the surface and interior of concrete at 180 days were taken for TG-DSC analysis. The results are shown in Figure 10. The absorption peak at about 400 °C is due to the decomposition of CH%, and the content of CH% on the surface and interior of the material was calculated.

The Table 3 shows the content of calcium hydroxide on the surface and interior of each sample. The results show that CH% on the surface decreased with increasing boric acid concentration, but the content inside, even for 70–180,000 ppm, presents a state of equivalent temperature. This point corresponds to the depth of its neutralization.

### 4.4. Pore Structure of Concrete

It can be seen from the Figure 11 that the total porosity of all specimens decreased as age increased. At the same age, the higher the concentration of boric acid, the larger the porosity of specimen. The porosity of 70–180,000 ppm was 1.43 times that of 70–0 ppm at 28 days. The probability of a macrovoid also increased as the concentration of boric acid increased. The total porosity decreased significantly at 20 and 40 °C at 90 days, and the pores with a diameter between 10 and 50 nm almost disappeared. The pores with three diameters also decreased at 90 days when compared those at with 28 days. However, the porosity of paste still increased as the concentration of boric acid increased. The total porosity of 70–180,000 ppm was 2.11 times of that of 70–0. At 180 days, the total porosity of 70–180,000 ppm was 1.59% more than that of 70–0 ppm. In particular, 70–18,000 ppm had more harmful pores, with a diameter greater than 50 nm, than the other test groups.

### 4.5. Scanning Electron Microscope

For further details, specimens at 28 days (early stage) and 180 days (late stage) under different reaction conditions were selected for SEM analysis, as shown in the following figures.

At 28 days, all specimens were at the early stage of reaction. In Figure 12, the morphological features of 20–50,000 ppm at 28 days were almost the same as those at 0 days. The set cement structure on the surface was obvious. The clear corrosion traces could be found on the surface of 40–90,000 ppm at 28 days.

The corroded pores were demonstrated on the surface, indicating that, in the early stage of reaction, paste was corroded by high-concentration boric acid even at a temperature of 40 °C for conservation. The surfaces of neither 70–0 nor 70–2000 ppm were obviously corroded at a temperature of 70 °C for conservation; however, many corrosion traces could be found on the surface of 70–7000 ppm, which was more seriously corroded than 40–90,000 ppm either in terms of corrosion area or corrosion depth. The corrosion area of 70–180,000 ppm further increased, and large pores emerged. The strong corrosion reaction occurred on the surface of 70–180,000 ppm at the early stage. It may be noticed that there were few set pores in the cement and that the internal section of paste was likely to be further damaged.

At 180 days, the corrosion of paste reached a late stage. In Figure 13, the corrosion on the surface was basically fixed. The surface of 20–50,000 ppm was still characterized by rich set cement. The corrosion area on the surface of 40–90,000 ppm at 180 days was greater than that at 28 days, but the surface was still dominated by a set cement structure.

At a temperature of 70 °C for conservation, the morphology on the surface of paste was further damaged as the concentration of boric acid increased. The obvious corrosion pores could be found on the surface of 70–2000 ppm. The paste material and reaction products started to be dissolved out from 70–2000 ppm, the corrosion area of which increased. Obvious dissolution traces could be found on the surface of 70–2000 ppm. No obvious set cement structure could be seen on the surface of 70–180,000 ppm, and its internal section was also obviously corroded.

### 4.6. Discussion

The neutralization of boric acid and the hydration of cement are like a rivalry for the calcium hydroxide in the paste. Therefore, when the concentration of boric acid is low, the hydration reaction in the paste plays a dominant role, and the properties of cement continuously increase.

However, when the concentration of boric acid is saturated at a temperature of 70 °C, the acid-base neutralization reaction plays a dominant role, and cement suffers from loss of properties during conservation. Therefore, the temperature of the reaction and the concentration of the acid solution have a significant influence on the neutralization and hydration of cement. The surface is firstly corroded by the neutralization between boric acid and cement, and then the internal section is gradually corroded. However, the hydration reaction occurs in all directions in paste. Even at 70 °C, the structure of cement could not be significantly destroyed by 180,000 ppm boric acid at the early stage.

Under the curing condition of high temperature and concentration, the corrosion mechanism of boric acid on cement is shown in Figure 14. The weight loss of 70–180,000 ppm concrete was not significantly changed when compared with other specimens between 0 and 28 days, but the 70–180,000 ppm paste started to corrode the interior between 28 and 90 days. The destruction of the properties of cement by the neutralization of boric acid was more serious than that by hydration. The properties of paste and concrete obviously declined at this age when compared with other test groups.

It is generally accepted that boric acid has little effect on concrete, but concrete shows a large performance loss at 70 °C and 180,000 ppm. These conditions are possible in nuclear power plants, so we cannot ignore the effects of boric acid leakage.

## 5. Conclusions

A batch of paste and concrete specimens were separately conserved in boric acid solutions of different concentrations at temperatures of 20, 40, and 70 °C. The conclusions are as follows:i.At the maximum allowable concentration of boric acid, 7000 ppm, even boric acid does not cause obvious performance loss to the outer wall of the pressurized water reactor at 20, 40, and 70 °C, and boric acid is safe for the external concrete structure of a pressurized water reactor under permissible operating conditions.ii.We simulated the 70 °C and 180,000 ppm scenario and found that, under a high-temperature and high-curing-concentration environment, the mass loss rate of 70–180,000 ppm was 0.69% at 180 days under conservation conditions, and its loss rate of compressive strength was 27.8% at 180 days. The surface corrosion of the concrete was serious. CH% in the surface was 3.12%, the neutralization depth was 1.5 mm, and the porosity of the harmful pores in the concrete surface was much higher than that of the rest of the specimens. The deterioration of the concrete surface caused by a high concentration of boric acid at a high temperature was the main reason for this loss of properties. Such a large loss rate of compressive strength is extremely dangerous in practical engineering.iii.Temperature and the concentration of boric acid are two important factors that cause concrete corrosion. The corrosion of concrete becomes more serious as the temperature and concentration of boric acid rise. However, at the early stage of the experiment, the hydration reaction of cement was stronger than the corrosion of boric acid at a high temperature, so the influence of the initial concentration was smaller than that of temperature. With the growth of the phase, the hydration reaction ended, and the influence of boric acid concentration began to dominate.iv.Ca(BO_2_)_2_ was detected on the surface of the 70–180,000 ppm concrete specimen at 180 days. This was not detected for any of the other specimens. As the concentration of boric acid increased, the porosity of harmful pores in the paste increased. The corrosion area on the surface of the paste gradually increased, along with corrosion depth. By combining this with the mass loss rate, it can be concluded that most of the substances produced during boric acid corrosion were dissolved into the acid conservation solution.v.Some highly active auxiliary cementites such as silica ash accelerate the hydration of cement, thus giving cement hydration in concrete an advantage over boric acid in the neutralization process. The addition of silica ash into cement is the focus of the next stage of our research.

## Figures and Tables

**Figure 1 materials-13-05036-f001:**
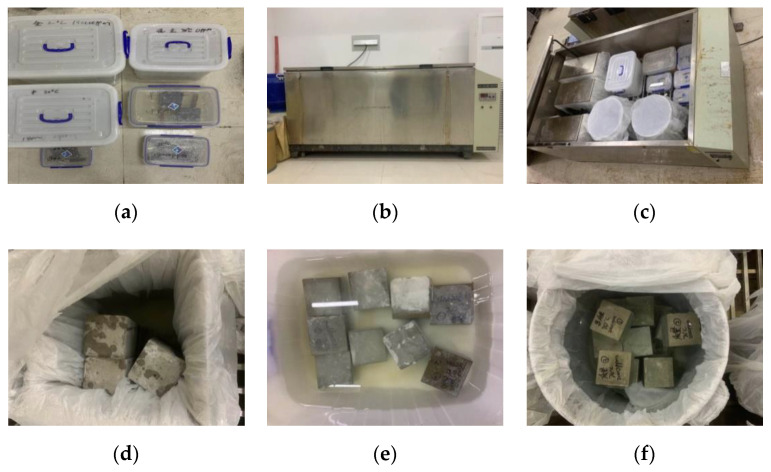
Process of boric acid curing. (**a**) Curing at 20 °C; (**b**,**c**) accelerated curing box; (**d**) curing at 40 °C; (**e**,**f**) curing at 70 °C.

**Figure 2 materials-13-05036-f002:**
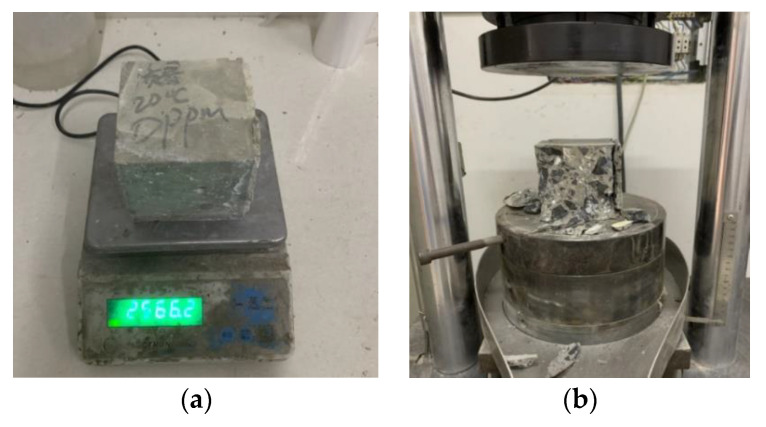
Mass compressive strength test on concrete. (**a**) electronic scaleand (**b**) electronic universal-testing machine.

**Figure 3 materials-13-05036-f003:**
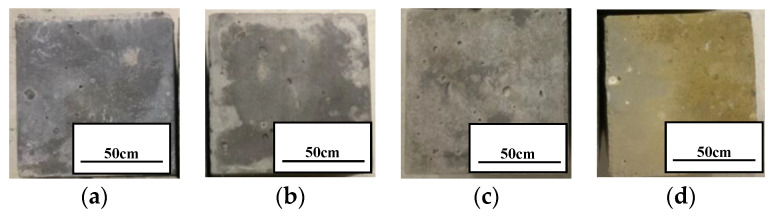
The morphology of concrete specimens at 70 °C and 90 days: (**a**) 0; (**b**) 2000; (**c**) 7000; (**d**) 180,000 ppm.

**Figure 4 materials-13-05036-f004:**
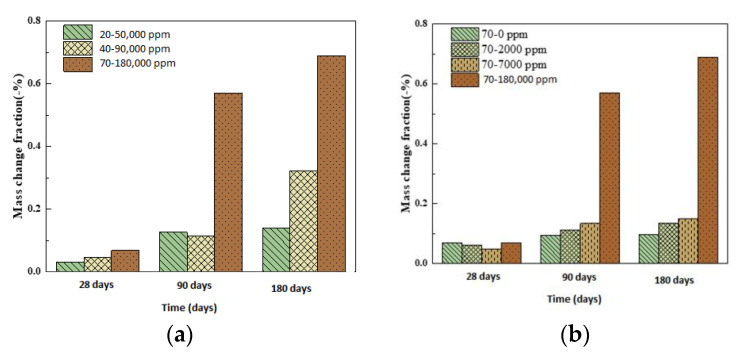
Mass loss of the concrete samples: (**a**) saturation concentration of boric acid; (**b**) 70 °C.

**Figure 5 materials-13-05036-f005:**
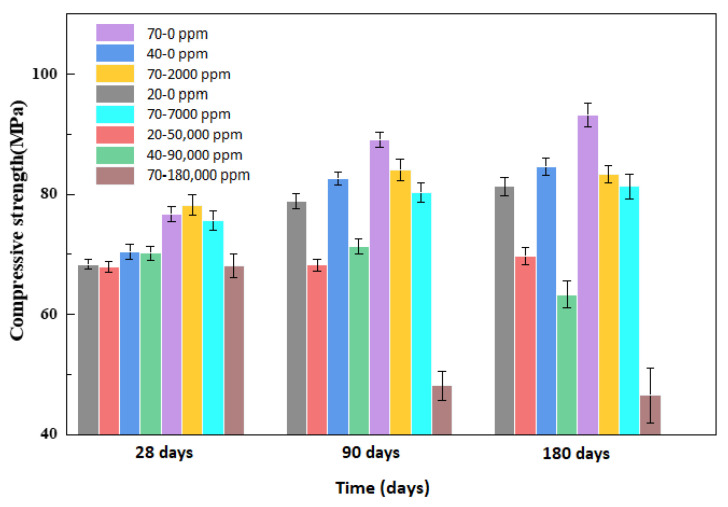
Compressive strength of the concrete samples at different ages.

**Figure 6 materials-13-05036-f006:**
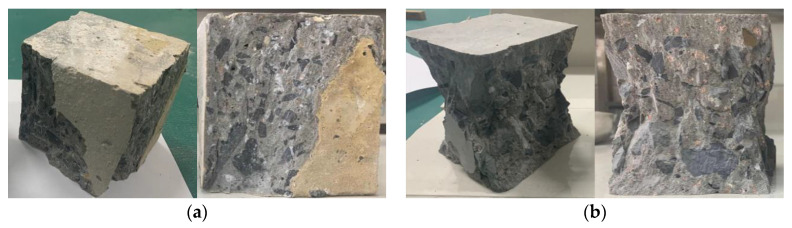
Appearance of 70–180,000 ppm concrete at 90 and 180 days. (**a**) 70–180,000 ppm; (**b**) other specimens.

**Figure 7 materials-13-05036-f007:**
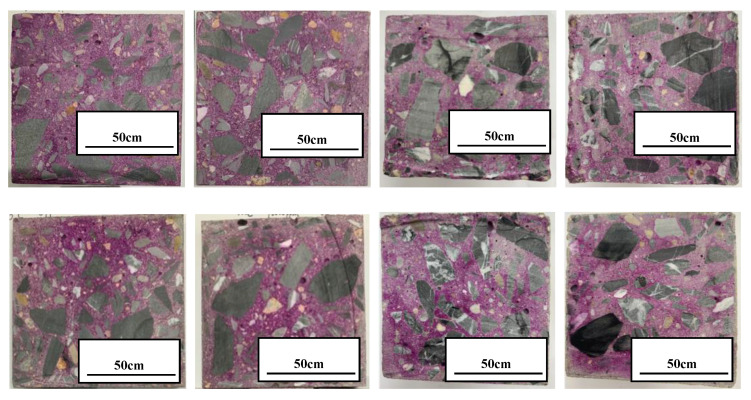
Depth of neutralization of concrete at 180 days.

**Figure 8 materials-13-05036-f008:**
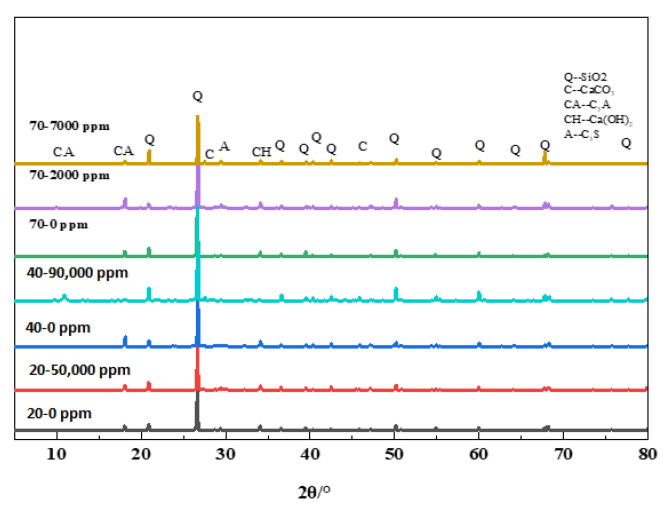
X-ray diffraction (XRD) patterns of concrete except 70–180,000 ppm at 180 days.

**Figure 9 materials-13-05036-f009:**
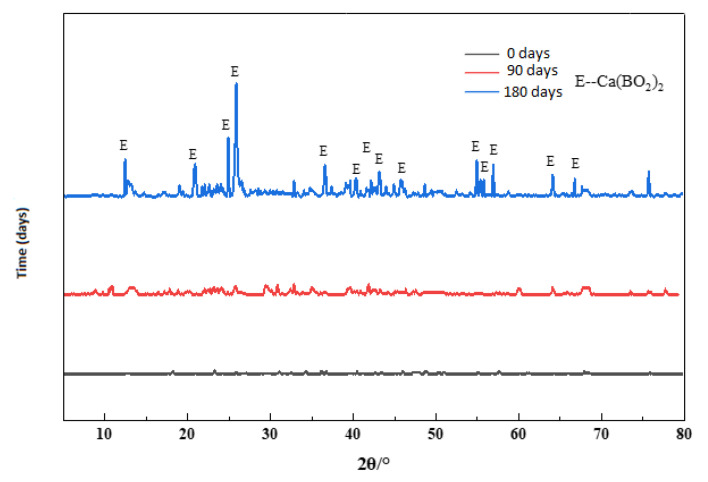
XRD patterns of 70–180,000 ppm at different ages (remove peaks of cement and sand).

**Figure 10 materials-13-05036-f010:**
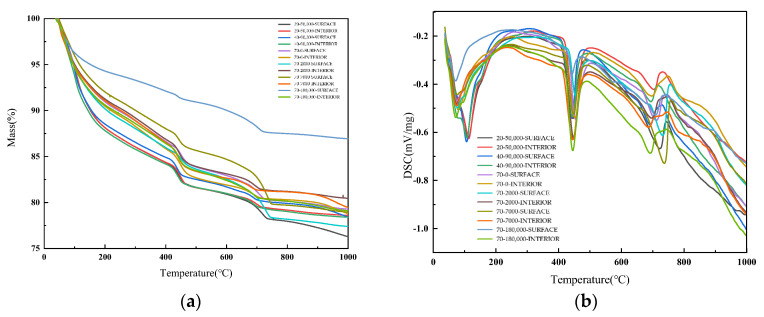
Thermogravimetry and differential scanning calorimetry (TG-DSC) curves of concrete at 180 days. (**a**) TG and (**b**) DSC.

**Figure 11 materials-13-05036-f011:**
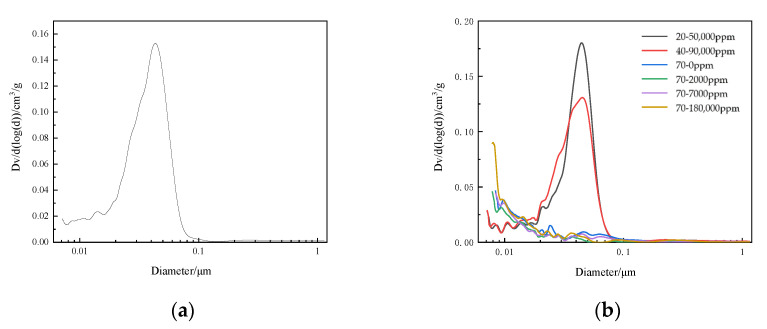
The porosity of the concrete at different ages. (**a**) 0 days; (**b**) 28 days; (**c**) 90d days and (**d**) 180 days.

**Figure 12 materials-13-05036-f012:**
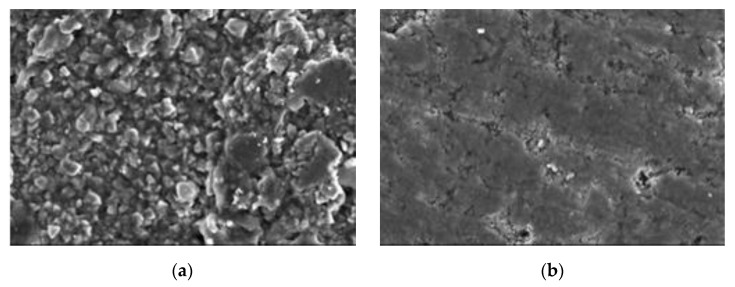
Typical SEM (1000 times) morphology images of cement paste at 28 days: (**a**) 20–50,000; (**b**) 40–90,000; (**c**) 70–0; (**d**) 70–2000; (**e**) 70–7000; and (**f**) 70–180,000 ppm.

**Figure 13 materials-13-05036-f013:**
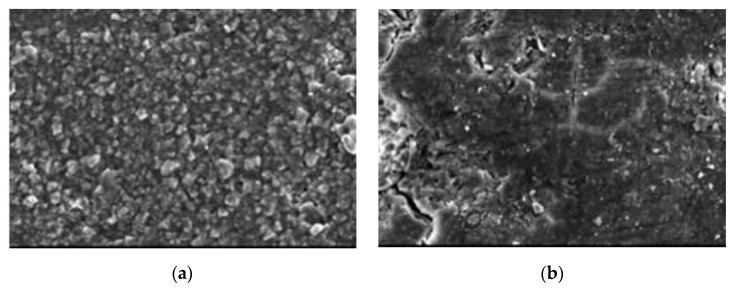
Typical SEM (1000 times) morphology images of cement paste at 180 days: (**a**) 20–50,000; (**b**) 40–90,000; (**c**) 70–0; (**d**) 70–2000; (**e**) 70–7000; and (**f**) 70–180,000 ppm.

**Figure 14 materials-13-05036-f014:**
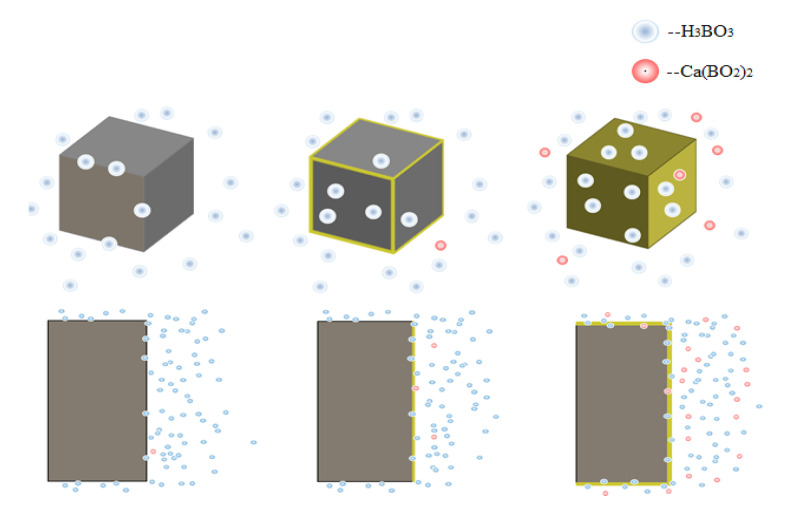
Corrosion mechanism of boric acid on concrete.

**Table 1 materials-13-05036-t001:** Chemical compositions of PC clinker and fly ash (FA).

Type	Chemical Compositions (%)							Loss on Ignition (%)
	SiO_2_	Al_2_O_3_	Fe_2_O_3_	CaO	MgO	SO_3_	R_2_O	
**PC**	19.43	4.73	2.96	64.00	2.35	2.58	0.71	2.81
**FA**	50.10	29.27	6.95	3.40	1.11	1.15	1.37	-

**Table 2 materials-13-05036-t002:** Concrete mix proportion (kg/m^3^).

Cement	Water	Fine Aggregate	Coarse Aggregate (5–16 mm)	Coarse Aggregate (16–31.5 mm)	Fly Ash	Water Reducer	Slump (mm)
415	160	688	538	538	62	5.7	165

**Table 3 materials-13-05036-t003:** The content of Ca (OH)_2_ at 180 days.

CH%	20–50,000	40–90,000	70-0	70–2000	70-7000	70–180,000
Surface	10.02	9.66	9.55	7.65	7.89	3.12
Interior	10.72	9.74	12.1	11.34	11.91	10.56

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
