# Peer review of "Effect of a Boric Acid Corrosive Environment on the Microstructure and Properties of Concrete"

_materials, 2020, doi:10.3390/ma13215036_

Round 1

Reviewer 1 Report

The manuscript presents an experimental work aimed at microstructure and properties of concrete cured with different concentrations of boric acid solution. There are following questions:

  1. The abstract needs to be revised and provided the important results from experiments or the main contribution of the manuscript.
  2. In Introduction, the literature review is too brief and should be improved, as should the concepts and conceptual scheme that supports the research that is not even referred to.
  3. What is the meaning of lines 85 to 86?
  4. Why are there different concentration designs for different curing temperatures? What is the basis?
  5. The author should explain how the boric acid concentration and temperature interact with the results of the study?
  6. The results of the study should be further analyzed and mechanism discussion, and compared with other studies.
  7. The author should explain the relationship between the micro results and the macro phenomenon.

Author Response

Dear reviewer:

Thank you for your Suggestions for modification. I have made corresponding modifications to the article. Now I submit the revised manuscript for your review.

Thank you very much.

Yours

Yu Wang

  1. The abstract needs to be revised and provided the important results from experiments or the main contribution of the manuscript.

The results show that the performance of specimens was stable under the curing conditions of 20 and 40  °C. Under the curing environment of 70 °C, the performance of concrete cured with 0, 2000, and 7000 ppm concentrations were stable, but the compressive strength of the 180,000 ppm specimen was reduced by 27.8% and suffered the most serious loss of mass and surface corrosion, with the most harmful pores. The high concentration of boric acid seriously damaged the surface structure of concrete, which is the main reason for its loss of properties. This situation is extremely dangerous in nuclear power engineering, so the effect of boric acid leakage cannot be ignored.

2.In Introduction, the literature review is too brief and should be improved, as should the concepts and conceptual scheme that supports the research that is not even referred to.

The deterioration of concrete is mainly due to the reduction of macroscopic mechanical properties, and the reason for the deterioration is the change of concrete structure. At present, the mechanism of deterioration of concrete is mainly analyzed by means of SEM and XRD [19, 20]. Since the pore structure is one of the main factors affecting the strength of concrete, it is very important to study the change of pores in concrete during the corrosion process [21].

3.What is the meaning of lines 85 to 86?

The concentrations of boric acid in the curing solution were 0 and 50,000 ppm at 20 °C; 0 and 90,000 ppm at 40 °C; and 0, 2000, 7000, and 180,000 ppm at 70 °C. These conditions are denoted by 20-0, 20-50,000, 40-0, 40-90,000, 70-0, 70-20,000, 70-7000, and 70-180,000 ppm for short.

4.Why are there different concentration designs for different curing temperatures? What is the basis?

The first thing I want to talk about is temperature, 20 ° C is the standard curing temperature, which is used for the control group; when the plant is operating normally, The temperature is usually about 40° C; 70 ° C is the maximum temperature allowed for nuclear power plants under normal circumstances.

20-0 ppm, 40-0 ppm and 70-0 ppm as control group too, 20-50000 ppm and 40-90000 ppm are the saturation concentration under the respective conditions. At 70 ° C, the concentration of boric acid during normal operation of the plant is 2,000 ppm, 7000 ppm is the maximum concentration allowed, 180000ppm is the saturation concentration.

5.The author should explain how the boric acid concentration and temperature interact with the results of the study?

Temperature and the concentration of boric acid are two important factors that cause concrete corrosion. The corrosion of concrete becomes more serious as the temperature and concentration of boric acid rise. However, at the early stage of the experiment, the hydration reaction of cement was stronger than the corrosion of boric acid at a high temperature, so the influence of the initial concentration was smaller than that of temperature. With the growth of the phase, the hydration reaction ended, and the influence of boric acid concentration began to dominate.

6.The results of the study should be further analyzed and mechanism discussion, and compared with other studies.

The neutralization of boric acid and the hydration of cement are like a rivalry for the calcium hydroxide in the paste. Therefore, when the concentration of boric acid is low, the hydration reaction in the paste plays a dominant role, and the properties of cement continuously increase.

Figure 12. Corrosion mechanism of boric acid on concrete.

However, when the concentration of boric acid is saturated at a temperature of 70 °C, the acid-base neutralization reaction plays a dominant role, and cement suffers from loss of properties during conservation. Therefore, the temperature of reaction and the concentration of acid solution have a significant influence on the neutralization and hydration of cement. The surface is firstly corroded by the neutralization between boric acid and cement, and then the internal section is gradually corroded. However, the hydration reaction occurs in all directions in paste. Even at 70 °C, the structure of cement could not be significantly destroyed by 180,000 ppm boric acid at the early stage.

Under the curing condition of high temperature and concentration, the corrosion mechanism of boric acid on cement is shown in Figure 12. The weight loss of 70-180,000 ppm concrete was not significantly changed when compared with other specimens between 0 and 28 days, but the 70-180,000 ppm paste started to corrode the interior between 28 and 90 days. The destruction of the properties of cement by the neutralization of boric acid was more serious than that by hydration. The properties of paste and concrete obviously declined at this age when compared with other test groups.

It is generally accepted that boric acid has little effect on concrete, but concrete shows a large performance loss at 70 °C and 180,000 ppm. These conditions are possible in nuclear power plants, so we cannot ignore the effects of boric acid leakage.

7.The author should explain the relationship between the micro results and the macro phenomenon.

We simulated the 70 °C and 180,000 ppm scenario and found that under a high-temperature and high-curing-concentration environment, the mass loss rate of 70-180,000 ppm was 0.69% at 180 days under conservation conditions, and its loss rate of compressive strength was 27.8% at 180 days. The surface corrosion of the concrete was serious. CH% in the surface was 3.12%, the neutralization depth was 1.5 mm, and the porosity of the harmful pores in the concrete surface was much higher than that of the rest of the specimens. The deterioration of the concrete surface caused by high concentration of boric acid at a high temperature was the main reason for this loss of properties. Such a large loss rate of compressive strength is extremely dangerous in practical engineering.

Reviewer 2 Report

The authors present a study on the effect of boric acid on the chemical stability and properties of concrete. Although the results presented are of interest, in my opinion, the paper has to be revised to meet the publication criteria. The following aspects have to be considered:

  1. On the scientific content:
    • It is unclear why the authors have chosen to cure samples in solutions containing boric acid. No comparison is made with samples exposed after normal curing (as it occurs for the real structures) thus, the conclusions are of questionable applicability.
    • The results presented in Figs. 4 and 5 shall incorporate standard deviation bars. How many replicas were employed?
    • Sections 2 and 3 shall be renamed or merged. In any case, the description of the testing experiment shall be improved. It is not clear how many of the faces of each cube are freely accessible to the aggressive solution. It seems that not all the 6 were.
    • The degradation process is an interfacial phenomenon (line 199) thus the corrosion rate results (lines 132 to 148) shall refer to the actual exposed surface.
  2. English usage/edition:
    • Line 39 “corrosion of boric acid” should be “corrosion by boric acid”.
    • Fig 1 caption. More explanatory details of each image are needed.
    • Fig 2 label. Reference to the mass determination is missing. In the text, it is also necessary to explain how mass was obtained, in humid od dry states.
    • Fig 6 label. Failure modes. More explanatory detail is needed.
    • Line 178 “I also” should be “we also”.
    • Line 213. Format of ref. 18.
    • Line 216. pH.
    • Line 234. The formula is missing.

---------------End of comments------------------------------------------------

Author Response

Dear reviewer:

Thank you for your Suggestions for modification. I have made corresponding modifications to the article. Now I submit the revised manuscript for your review.

Thank you very much.

Yours

Yu Wang

  1. On the scientific content:
  • It is unclear why the authors have chosen to cure samples in solutions containing boric acid. No comparison is made with samples exposed after normal curing (as it occurs for the real structures) thus, the conclusions are of questionable applicability.

Samples exposed after normal curing is a common situation in nuclear power plants. But that means there's no leakage of boric acid, and it doesn't affect the concrete structure on the outer wall. This study focuses on the situation after boric acid leakage, so all specimens were cured in solution.

  • The results presented in Figs. 4 and 5 shall incorporate standard deviation bars. How many replicas were employed?

The compressive strength is measured  according to GB/T 50107-2010: Standard for evaluation of concrete compressive strength, and each set of data has three specimens.A line chart was drawn to give you a more intuitive view of the rate of change in strength, but this is not standard to modify.

  • Sections 2 and 3 shall be renamed or merged. In any case, the description of the testing experiment shall be improved. It is not clear how many of the faces of each cube are freely accessible to the aggressive solution. It seems that not all the 6 were.

       The title of section 2 has been changed.

       The corrosion of boric acid on concrete is relatively average. During curing, all six surfaces are immersed in boric acid solution, so each sampling is a typical sample, and at least three groups of samples are selected. The experimental data under the same reaction conditions are almost the same, including the bottom surface that the specimen contacts with the curing box.

  • The degradation process is an interfacial phenomenon (line 199) thus the corrosion rate results (lines 132 to 148) shall refer to the actual exposed surface.

Yes, boric acid corrosion has only damaged the concrete surface at 180d, the quality loss of concrete come from its surface.

   2.English usage/edition:

  • Line 39 “corrosion of boric acid” should be “corrosion by boric acid”.

I have modified it.

  • Fig 1 caption. More explanatory details of each image are needed.

Add “Figure 1. situation of boric acid curing

(a) curing at 20 ℃  (b,c) accelerated curing box  (d) curing at 40 ℃ (e,f)curing at 70 ℃

  • Fig 2 label. Reference to the mass determination is missing. In the text, it is also necessary to explain how mass was obtained, in humid od dry states.

I've increased the scale. The specimen was taken out of the curing box and dried for the same drying time for each age.

  • Fig 6 label. Failure modes. More explanatory detail is needed.

Add” (a) 70-180000ppm  (b) other specimens”

  • Line 178 “I also” should be “we also”.

I have modified it.

  • Line 213. Format of ref. 18.

I have modified it.

Niu, J.; Niu, D. Experimental study on the neutralization of fly ash concrete attacked by acid rain, 2011 International Conference on Electric Technology and Civil Engineering, Lushan, China, 22-24 April, 2011; pp 3162-3165. [CrossRef]

  • Line 216. pH.

I have modified it.

  • Line 234. The formula is missing.

       The sentence was deleted

Reviewer 3 Report

Please find attached a PDF file with my comments and suggestions for authors.

Author Response

Dear reviewer:

Thank you for your Suggestions for modification. I have made corresponding modifications to the article. Now I submit the revised manuscript for your review.

Thank you very much.

Yours 

Yu Wang

Firstly, in relation to the introduction section, the state‐of‐art review included here is enough, with 18 references cited. However, in my opinion the aim and the novelty of the research work included in the paper are not clearly explained in this section. I suggest to add a final paragraph in the introduction section with a explicit description of the objectives and the innovation of this work.

In the existing research, only the effect of boric acid on cement concrete at 20 °C has been studied, without taking into account the effect of temperature. Therefore, this study used the actual situation of a nuclear power plant as a simulation and the concentration of boric acid was set considering the polycondensation of boric acid in the limit case (70 °C). Thus, a batch of concrete specimens was formed, and an accelerated curing box was used to simulate the cooling water tank with curing temperatures of 40 and 70 °C. Concrete specimens cured with saturated acid concentrations were used to study the effect of boric acid on the microstructure and properties of cement concrete materials.

In the section 2, the materials studied are well explained. Nevertheless, I have found a contradiction in the names of the sections 2 and 3. The section 2 is named “Materials and Methods” and section 3 is named “Methods”. Therefore, I suggest to re‐name section 2 as “Materials” only, or merge both sections in one called “Materials and Methods”. In the current section 2, the table 2 must be included in the same page (now it appears divided in pages 2 and 3).

I have modified it.

In relation to section 3, the caption of the section 2 must be included in the same page than the figure. In lines 95 and 96, the standard followed for determining the compressive strength must be indicated and cited.

The compressive strength is measured  according to GB/T 50107-2010: Standard for evaluation of concrete compressive strength, and each set of data has three specimens.

In subsection 3.3, it would be interesting to indicate the equipment (model and manufactured) used for performing the XRD and TG‐DSC analysis. I do not understand the sentence “and analyzed the corrosion process of the material from the microscopic perspective” (lines 116‐117) when you are describing the MIP technique.

The sentence has been modified to” and analyzed the corrosion process of the material from the pore structure.”

Please rewrite this sentence o delete it. In addition to this, I do not understand the text in lines 119‐121, so I would be good that you improve that explanation about the application of SEM in this research.

In order to investigate the morphology of concrete under boric acid attack, a characteristic fresh piece of concrete was sampled, hydration terminated, and vacuum dried,  Then, the dried sample was embedded in a low-modulus epoxy, coated with a gold-palladium coating, SEM (JSM-6510, JEOL, Japan) was used to observe the surface morphology of the concrete.

With respect to the section “4 Results and discussion”, the results are well described, and hardly discussion is included. The current subsection 4.6 “Discussion” is very poor and short, and there is no references cited here, when they are necessary to support your arguments. I think that the experimental setup conducted in this research and the results obtained are wide, and they do

deserve a poor discussion like this. The discussion is the most important part of a scientific manuscript, so without a good discussion in my opinion the manuscript must not be accepted for publication. A high improvement is needed in this line.

Regarding the conclusion section, I think that the bullet points must be summarized in order to make clearer the relevant findings of the research included in the manuscript.

  • At the maximum allowable concentration of boric acid, 7000 ppm, even boric acid does not cause obvious performance loss to the outer wall of the pressurized water reactor at 20, 40, and 70 °C, and the boric acid is safe for the external concrete structure of a pressurized water reactor under permissible operating conditions.
  • We simulated the 70 °C and 180,000 ppm scenario and found that under a high-temperature and high-curing-concentration environment, the mass loss rate of 70-180,000 ppm was 0.69% at 180 days under conservation conditions, and its loss rate of compressive strength was 27.8% at 180 days. The surface corrosion of the concrete was serious. CH% in the surface was 3.12%, the neutralization depth was 1.5 mm, and the porosity of the harmful pores in the concrete surface was much higher than that of the rest of the specimens. The deterioration of the concrete surface caused by high concentration of boric acid at a high temperature was the main reason for this loss of properties. Such a large loss rate of compressive strength is extremely dangerous in practical engineering.
  • Temperature and the concentration of boric acid are two important factors that cause concrete corrosion. The corrosion of concrete becomes more serious as the temperature and concentration of boric acid rise. However, at the early stage of the experiment, the hydration reaction of cement was stronger than the corrosion of boric acid at a high temperature, so the influence of the initial concentration was smaller than that of temperature. With the growth of the phase, the hydration reaction ended, and the influence of boric acid concentration began to dominate.
  • Ca(BO2)2 was detected on the surface of the 70-180,000 ppm concrete specimen at 180 days. This was not detected for any of the other specimens. As the concentration of boric acid increased, the porosity of harmful pores in the paste increased. The corrosion area on the surface of the paste gradually increased, along with corrosion depth. By combining this with the mass loss rate, it can be concluded that most of the substances produced during boric acid corrosion were dissolved into the acid conservation solution.
  • Some highly active auxiliary cementites such as silica ash accelerate the hydration of cement, thus giving cement hydration in concrete an advantage over boric acid in the neutralization process. The addition of silica ash into cement is the focus of the next stage of our research.

Reviewer 4 Report

This paper  determines the attack of  boric acid at 70*C on hardened concrete. This attack simulates  conditions in  certain  types of nuclear  reactors. It is thus of interest to a  small segment  of  scientists and  engineers  but potentially valuable  to safety engineers. It thus  is appropriate  for consideration.

The  experimental programme is  generally appropriate and  well resourced to use  modern methods.. However it suffers from an number iof defects  some  serious

I deal with general points  first

The attack is  depicted as a neutralisation attack. But no pH values are given, and more seriously,  no analysis of the  aqueous solution are presented. Thus the composition of the attacking solution   may have decreased in boron content during exposure so that the actual concentration was  much less than th calculated  composition.   The  authors are simulating attack where the  boric acid concentration is constant  but  this  constancy of composition cannot be assured for their simulants. If  the concentration  depletes rapidly, their conclusions- which assume  constant composition- are invalid.  So this point MUST be addressed.

I have  number of other comments

  • BEFORE Abstract. This will be widely read and must be clear. It must be stated that the concrete was  made and cured  BEFORE   testing  in boric acid. This is to avoid giving the impression that boric  acid was added to fresh cements . The  term “cure” should be reserved  for  the period   allowed for ,
  • This is too long. All the  information  could be condensed into a few sentences.
  • Materials and methods. What were the humidity and temperature in the curing stage?
  • Figures 1 nan 2 ack a sxale and do not contribute to the Delete.
  • Compositions of the aqueous phase ae often defined by two values  with a hyphen. Is this shorthand for all compositions within that range?. Please define.
  • Fig 3 lacks a scale. .
  • I am not convinced that the surface visual colour change is important. What is its significance? Does it need a figure?. And are the shanges  visible after drying?
  • Are measurements made at ambient Fig 5 and associated text. How many samples were tested at each point and what is the standard  deviation of  strength values? . Was measure ment done at ambient following cure?. Were samples dried before  compressive  strengths were determined?
  • Figure 7 lacks a scale and it is noteasy to see the boundary   between leached and unleached. I ecommend deletion of the figure and  instead provide a  Table showing measured 
  • Perhaps the wakest section is that on the X ray diffraction. Two theta values are useless unless the radiation is also      Identification of some phases seemst o rest on  a single  reflection and this  low standar of proof is not acceptable. The sentence  (line 211) is unclear. Wht  was unexpeted  and why?
  • How was sampling done for the data in Table 9?    Was some  averaging done?
  • The paper contains two figures both numbered “Fig 10”.

Figures 10 and 11 lack scale. What do they show?  Are we looking at surfaces?  . I am not sure  what  I am to conclude from  the figures. If the amswer is “nothing”  they should  be deleted.. .

  • Summary the paper contains  useful information but is not acceptable in its present form/ Moreover its conclusions  need to be  qualified unless it boric  acid concentration remained constsnt during  the  immersion

Author Response

Dear reviewer:

Thank you for your Suggestions for modification. I have made corresponding modifications to the article. Now I submit the revised manuscript for your review.

Thank you very much.

Yours 

Yu Wang

The attack is  depicted as a neutralisation attack. But no pH values are given, and more seriously,  no analysis of the  aqueous solution are presented. Thus the composition of the attacking solution   may have decreased in boron content during exposure so that the actual concentration was  much less than th calculated  composition.   The  authors are simulating attack where the  boric acid concentration is constant  but  this  constancy of composition cannot be assured for their simulants. If  the concentration  depletes rapidly, their conclusions- which assume  constant composition- are invalid.  So this point MUST be addressed.

We have analyzed the curing solution.The pH of the solution of boric acid at 70℃ (180000ppm) was about 4.2, and the pH of the solution did not change significantly within a week, mannitol was used to measure concentration.so, we changed the curing solution every week to ensure the concentration of boric acid, this point has been mentioned in the article.

I have  number of other comments

  • BEFORE Abstract. This will be widely read and must be clear. It must be stated that the concrete was  made and cured  BEFORE   testing  in boric acid. This is to avoid giving the impression that boric  acid was added to fresh cements . The  term “cure” should be reserved  for  the period   allowed for ,

The title has been changed to” Effect of boric acid corrosive environmenton microstructure and properties of concrete”.

  • This is too long. All the  information  could be condensed into a few sentences.

For those majoring in materials, some descriptions of the article are really not necessary. However, there will be some admirers who will read the articles of this journal. Therefore, Therefore, I will try my best to describe them in detail so that some non-majors can also read the articles well, and I hope that reviewers and teachers can understand them.

  • Materials and methods. What were the humidity and temperature in the curing stage?

All specimens were cured at 20℃ for 28 days.(February 3 - March 3).The specimens were placed in a curing box with a humidity of 95% and a temperature of 25℃.

  • Figures 1 nan 2 ack a sxale and do not contribute to the Delete.

Some experts asked for proof that the experiment was real, so I had to add some pictures to prove it.

  • Compositions of the aqueous phase ae often defined by two values  with a hyphen. Is this shorthand for all compositions within that range?. Please define.

A concrete accelerated curing box was used to simulate the actual temperature. In this experiment, we used 20, 40, and 70 °C as a comparative test. The concentrations of boric acid in the curing solution were 0 and 50,000 ppm at 20 °C; 0 and 90,000 ppm at 40 °C; and 0, 2000, 7000, and 180,000 ppm at 70 °C. These conditions are denoted by 20-0, 20-50,000, 40-0, 40-90,000, 70-0, 70-20,000, 70-7000, and 70-180,000 ppm for short.

  • Fig 3 lacks a scale. .

I added a scale.

  • I am not convinced that the surface visual colour change is important. What is its significance? Does it need a figure?. And are the shanges  visible after drying?

The yellowing of the concrete surface indicates that the cement on the concrete surface has been eroded away, exposing the sand inside

The color change indicates that the concrete will gradually decrease with the increase of boric acid concentration, which corresponds to the loss of mass.

  • Are measurements made at ambient Fig 5 and associated text. How many samples were tested at each point and what is the standard  deviation of  strength values? . Was measure ment done at ambient following cure?. Were samples dried before  compressive  strengths were determined?

The compressive strength is measured  according to GB/T 50107-2010: Standard for evaluation of concrete compressive strength, and each set of data has three specimens. The specimen was measured after drying

  • Figure 7 lacks a scale and it is noteasy to see the boundary   between leached and unleached. I ecommend deletion of the figure and  instead provide a  Table showing measured 

I added a scale. Since the neutralization depth of most specimens is 0, the table is not used, but a more intuitive depth of 70-180,000.

  • Perhaps the wakest section is that on the X ray diffraction. Two theta values are useless unless the radiation is also      Identification of some phases seemst o rest on  a single  reflection and this  low standar of proof is not acceptable. The sentence  (line 211) is unclear. Wht  was unexpeted  and why?

Boric acid on the corrosion of concrete is relatively flat, only under the condition of 70-180000 is strong, 70-180000 surface uneven, the resulting in the loss of the cement at the same time also will be the loss of reactants, so the content of reactants is less, only in the 180 d is measured by partial boric acid calcium, and calcium metaborate as the position of the peak and documented come, so will its see as metaborate calcium.

  • How was sampling done for the data in Table 9?    Was some  averaging done?

The corrosion of boric acid on concrete is relatively average. During curing, all six surfaces are immersed in boric acid solution, so each sampling is a typical sample, and at least three groups of samples are selected. The experimental data under the same reaction conditions are almost the same, including the bottom surface that the specimen contacts with the curing box. Three samples were selected for each data of this set of results, and the results of the three samples were similar.

  • The paper contains two figures both numbered “Fig 10”.

I have modified it.

Figures 10 and 11 lack scale. What do they show?  Are we looking at surfaces?  . I am not sure  what  I am to conclude from  the figures. If the amswer is “nothing”  they should  be deleted.. .

I have modified it. There is a scale, but the picture is too small. I enlarged the picture so that I could see the scale. The sample is the surface of concrete.

This proves that with the increase of boric acid concentration, the surface corrosion of the specimen becomes more serious.As well as surface roughness to demonstrate the loss of reactants.

  • Summary the paper contains  useful information but is not acceptable in its present form/ Moreover its conclusions  need to be  qualified unless it boric  acid concentration remained constsnt during  the  immersion

Mannitol was used to measure concentrationso, we changed the curing solution every other week to ensure the concentration of boric acid, this point has been mentioned in the article.

Round 2

Reviewer 1 Report

I have no other comments.

Author Response

Dear reviewer:

Thank you for your comments and suggestions of my paper, which will become a milestone in my scientific research career. I will work harder for this.

My best wishes.

Yours

Yu Wang

Reviewer 3 Report

The majority of my comments has been addressed and the manuscript has been greatly improved during the review process. In my opinion, the manuscript can be accepted for publication.

Author Response

(The authors gave the same response as above.)

Reviewer 4 Report

The manuscript has many sections in yellow high;ight and I am unsure what this  highlighting means. 

This is a second review.  The authors  have addressed  the comments  of the first review in the most superficial manner as possible. For example , I  commented in  my first review  that the  X ray evidence  was insufficient to  prove the presence of calcium borate. Ons  small, almost invisible diffraction blip. was not proof. I spent  a lot of time   on the first review and if the authors are going to content themselves  with  seeing  how little they can change,  my  efforts to improve the paper are wasted. I cannot  accept the  changes as adequate,  

Author Response

Dear reviewer:

Thank you for your comments and suggestions of my paper. Your suggestion is very interesting and practical.I thank you for that and have modified the XRD , removed the peaks of cement and sand, and submitted the article to a professional organization for English language service.

I hope my article can get your support, which will become a milestone in my scientific research career and inspire me to strive for scientific research all my life

My best best best wishes.

Yours

Yu Wang
